# Purine Metabolism and Hexosamine Biosynthetic Pathway Abnormalities in Diarrheal Weaned Piglets Identified Using Metabolomics

**DOI:** 10.3390/ani14030522

**Published:** 2024-02-05

**Authors:** Mingyu Wang, Qin Zhong, Huailu Xin, Bing Yu, Jun He, Jie Yu, Xiangbing Mao, Zhiqing Huang, Yuheng Luo, Junqiu Luo, Hui Yan, Aimin Wu, Junning Pu, Ping Zheng

**Affiliations:** Key Laboratory for Animal Disease-Resistance Nutrition of Sichuan Province, Animal Nutrition Institute, Sichuan Agricultural University, Chengdu 611130, China; wmy970602@163.com (M.W.); 15207121708@163.com (Q.Z.); 18782253320@163.com (H.X.); ybingtian@163.com (B.Y.); hejun8067@163.com (J.H.); jerryyujie@163.com (J.Y.); acatmxb2003@163.com (X.M.); luoluo212@126.com (Y.L.); 13910@sicau.edu.cn (J.L.); yan.hui@sicau.edu.cn (H.Y.); wuaimin0608@163.com (A.W.); junningpu@163.com (J.P.)

**Keywords:** post-weaning diarrhea, metabolomic analysis, piglets, stress

## Abstract

**Simple Summary:**

Post-weaning diarrhea significantly impacts the survival and growth performance of piglets. This study aimed to obtain further insights into the metabolic changes in post-weaning diarrhea. The results showed that purine metabolism and hexosamine biosynthetic pathway were disturbed in diarrheal weaned piglets. This study suggests further studies could focus on dietary strategies, including energy strategies, ultimately alleviate diarrhea in weaning piglets via purine metabolism and hexosamine biosynthetic pathway.

**Abstract:**

Post-weaning diarrhea significantly contributes to the high mortality in pig production, but the metabolic changes in weaned piglets with diarrhea remain unclear. This study aimed to identify the differential metabolites in the urine of diarrheal weaned piglets and those of healthy weaned piglets to reveal the metabolic changes associated with diarrhea in weaned piglets. Nine 25-day-old piglets with diarrhea scores above 16 and an average body weight of 5.41 ± 0.18 kg were selected for the diarrhea group. Corresponding to the body weight and sex of the diarrhea group, nine 25-month-old healthy piglets with similar sex and body weights of 5.49 ± 0.21 kg were selected as the control group. Results showed that the serum C-reactive protein and cortisol of piglets in the diarrhea group were higher than those in the control group (*p* < 0.05). The mRNA expression of *TNF-α*, *IFN-γ* in the jejunum and colon, and *IL-1β* in the jejunum were increased in diarrhea piglets (*p* < 0.05), accompanied by a reduction in the mRNA expression of *ZO-1*, *ZO-2*, and *CLDN1* in the jejunum and colon (*p* < 0.05); mRNA expression of *OCLN* in the colon also occurred (*p* < 0.05). Metabolomic analysis of urine revealed increased levels of inosine, hypoxanthine, guanosine, deoxyinosin, glucosamine, glucosamine-1-p, N-Acetylmannosamine, chitobiose, and uric acid, identified as differential metabolites in diarrhea piglets compared to the controls. In summary, elevated weaning stress and inflammatory disease were associated with the abnormalities of purine metabolism and the hexosamine biosynthetic pathway of weaned piglets. This study additionally indicated the presence of energy metabolism-related diseases in diarrheal weaned piglets.

## 1. Introduction

Piglets experience stress throughout the weaning process when solid feed takes the place of maternal milk as their primary source of nutrition, accompanied by environmental changes [1,2]. Solid feed is difficult to metabolize in early-stage piglets due to inadequate secretion of digestive enzymes [3,4]. This results in the disruption of the intestinal physical barrier, including tight junctions. Damage to tight junctions leads to the permeation of pathogenic microorganisms into the basement membrane and other tissues, triggering enteric infections [5,6].

Weaning stress could result in immune dysfunction, decreased appetite, and digestive problems [1,7]. Therefore, weaned piglets are susceptible to pathological diarrhea, characterized by increased defecation frequency and water content in their feces. Studies have shown that weaned piglets usually suffer from inflammatory disease in their intestine and liver [8,9,10,11,12]. These inflammatory diseases are significant contributors to the high mortality and worse growth performance associated with post-weaning diarrhea [13]. In the past, intensive pig farms primarily relied on antibiotics as the primary strategy to mitigate post-weaning diarrhea [14]. Currently, reducing antibiotic use is urgent due to the harm of antibiotic resistance and residues to human and animal health [15,16,17,18]. However, reducing antibiotic utilization exacerbates the inflammatory diseases in weaned piglets living in an environment filled with pathogenic microorganisms, further escalating the threats of post-weaning diarrhea to piglet development. Therefore, it is necessary to identify metabolic pathological alterations in diarrheal weaned piglets to establish a foundation for future research on effective nutritional interventions to alleviate post-weaning diarrhea in piglets.

Metabolic changes are directly related to pathological physiology, and metabolomics has been proven effective in identifying specific biomarkers for organ damage [19]. This study employed metabolomic techniques to identify the disparities in urine metabolites between diarrheal and healthy weaned piglets. This study aimed to obtain further insights into weaning stress and inflammatory disease in weaned piglets at the metabolic level. This will provide significant information for determining nutritional approaches to reduce post-weaning diarrhea.

## 2. Materials and Methods

### 2.1. Experimental Design

The study was approved by the Sichuan Agricultural University animal welfare committee and carried out in accordance with the National Research Council’s Guide for the Care and Use of Laboratory Animals. All experimental piglets in this study are available at the Teaching and research base of Sichuan Agricultural University. Duroc × Landrace × Yorkshire (DLY) Piglets were weighed at 21-day-old upon weaning. Diarrhea scores were recorded twice a day (morning and evening) from the first to the fourth day post-weaning, according to the criteria outlined in Appendix A. In brief, piglets that excreted hard bar/hard granulous feces were given a score of 0, those that excreted soft/forming feces were given a score of 1, those that excreted dense/non-formed feces were given a score of 2, and those that excreted fluid/non-formed feces were given a score of 3. Piglets with a total diarrhea score of ≤4 were classified as healthy, while those with a score ≥ 16 were considered to be diarrheal piglets. Nine piglets with similar post-weaning weights of 5.41 ± 0.18 kg and with diarrhea scores above 16 were selected from a total of 135 piglets in 12 litters and were marked as the diarrheal group (DIA group, *n* = 9). Subsequently, nine healthy piglets with similar post-weaning body weights of 5.49 ± 0.21 kg and a similar sex ratio to the DIA group were selected and marked as the control group (CT group, *n* = 9). The post-weaning weights of piglets in the DIA and CT groups are shown in Appendix A. On the fifth day post-weaning, piglets were weighed and blood samples were collected via the anterior vena cava. Subsequently, the piglets were anesthetized and slaughtered; samples of intestinal mucosa were collected for further parameter analysis.

### 2.2. Sample Collection

Anticoagulant-free vacuum tubes were used to collect blood samples from the anterior vena cava. The blood samples were, thereafter, subjected to centrifugation at a speed of 3500 revolutions per minute for a duration of 15 min at a temperature of 4 degrees Celsius. The resulting serum was then separated and preserved at a temperature of −20 degrees Celsius for subsequent examination [20]. Subsequently, the piglets were executed using an intramuscular administration of pentobarbital sodium at a dosage of 3 mg per kilogram of body weight. Immediately after, the abdominal cavity was opened to extract the jejunum and colon. The segments of the intestine were extracted, dissected lengthwise, and rinsed with physiological saline at a low temperature. The intestinal mucosa was obtained by gently scraping it with a sterile glass slide. The collected mucosa was then rapidly frozen in liquid nitrogen and stored at a temperature of −80 °C until it was ready for further examination [20]. After slaughter, the urinary bladder of each piglet was excised; urine was extracted from the urinary bladder using a syringe for metabolomic analysis.

### 2.3. Profiling of Serum Samples

Serum interleukin 18 (IL-18), IL-1β, and tumor necrosis factor α (TNF-α) were measured using the corresponding ELISA kits (Jiangsu Meimian Industrial Co., Ltd., Yancheng, China) according to the manufacturer’s instructions. The levels of serum C-reactive protein and cortisol were determined using a fully automated biochemical analyzer (Biochemical Analytical Instrument, Beckman CX4, Beckman Coulter Inc., Brea, CA, USA).

### 2.4. Quantitative Real-Time Polymerase Chain Reaction (qRT-PCR)

The extraction steps for total RNA were referenced from previous studies [20]. In brief, TRIzol Reagent (TaKaRa, Dalian, China) was used to isolate intestinal mucosa total RNA per the manufacturer’s instructions. The concentration and purity of total RNA were measured using a Beckman Coulter DU800 spectrophotometer at 260 and 280 nm. The 260/280 nm absorption ratio was 1.8–2.0 for all samples. Reverse transcription was performed for each sample using the Prime Script RT Reagent Kit with gDNA Eraser (TaKaRa, Dalian, China) per the manufacturer’s instructions [20].

A quantitative real-time polymerase chain reaction (PCR) was performed to analyze the expression levels of the tumor necrosis factor α (*TNF-α*), interleukin 1*β* (*IL-1β*), interferon γ (*IFN-γ*), zonula occludens-1 (*ZO-1*), *ZO-2*, occludin (*OCLN*), and claudin 1 (*CLDN1*) using SYBR Premix Ex Taq II (Tli RNaseH Plus) reagents (TaKaRa, Dalian, China) and the QuanStudio 6 Flex Real-Time PCR detection system (Applied Biosystems, Foster City, CA, USA). All primers were commercially synthesized and purified by Sangon Biotech Co., Ltd. (Shanghai, China) and were shown in Table 1. The reaction conditions were referenced from previous studies [20,21]. In brief, the reaction was performed in 10 μL consisting of 5 μL of SYBR Premix Ex Taq (2×, TaKaRa Biotechnology, Dalian, China), 1 μL each of reverse and forward primer, 2 μL of double-distilled water, and 1 μL of cDNA template. The thermal cycling parameters were as follows: 90 °C for 30 s and 40 cycles of 95 °C for 10 s, 60 °C for 25 s, and 72 °C for 5 min. To ensure reaction specificity, each real-time quantitative PCR test underwent melting curve analysis after amplification. The reference gene β actin was used to normalize the mRNA expression of target genes. Target gene mRNA expression levels were determined using the 2^−ΔΔCt^ technique [20,21].

### 2.5. Chromatography Conditions

Samples were retrieved from −80° C storage, gradually dissolved at a temperature of 4 °C, and then 100 μL of each sample from each group was individually extracted. Afterwards, 400 μL of methanol–acetonitrile solution (1:1, *v*/*v*) that had been pre-chilled was added, and the mixture was vigorously mixed for 60 s using a vortex. The samples were placed at −20 °C for 1 h to precipitate the proteins, followed by centrifugation at 14,000 rcf for 20 min at 4 °C. The supernatant was collected, freeze-dried, and stored at −80 °C for subsequent analysis.

The samples were separated using an Agilent 1290 Infinity LC Ultra-High-Performance Liquid Chromatography (UHPLC) system with a HILIC column. The column temperature was maintained at 25 °C, and the flow rate was set at 0.3 mL/min. The mobile phase composition was as follows: A—water + 25 mM ammonium acetate + 25 mM ammonia solution; B—acetonitrile. The gradient elution program was as follows: 0 to 0.5 min, 95% B; 0.5 to 7 min, a linear decrease of B from 95% to 65%; 7 to 8 min, a linear decrease of B from 65% to 40%; 8 to 9 min, B maintained at 40%; 9 to 9.1 min, a linear increase of B from 40% to 95%; 9.1 to 12 min, B maintained at 95%. Throughout the analysis, samples were kept in a 4 °C autosampler to prevent fluctuations in instrument detection signals. To minimize the impact of instrument signal fluctuations, samples were analyzed in a random order. Quality control samples were inserted into the sample queue to monitor and assess the stability of the system and the reliability of experimental data.

### 2.6. Quadrupole Time-of-Flight Mass Spectrometry Conditions

The detection was performed using electrospray ionization (ESI) in positive and negative ion modes, respectively. After sample separation via UHPLC, mass spectrometric analysis was carried out using an Agilent 6550 mass spectrometer. The ESI source conditions were set as follows: Gas Temperature: 250 °C, Drying Gas: 16 L/min, Nebulizer: 20 psig, Sheath Gas Temperature: 400 °C, Sheath Gas Flow: 12 L/min, Vcap: 3000 V, Nozzle Voltage: 0 V. Fragment: 175 V, Mass Range: 50–1200, Acquisition Rate: 4 Hz, Cycle Time: 250 ms. Following sample detection, the AB Triple TOF 6600 mass spectrometer was employed for metabolite identification, and the primary and secondary spectra of QC samples were collected. The ESI source conditions for this instrument were as follows: Ion Source Gas1 (Gas1): 40, Ion Source Gas2 (Gas2): 80, Curtain Gas (CUR): 30, Source Temperature: 650 °C, IonSapary Voltage Floating (ISVF) ±5000 V (positive and negative modes). The secondary mass spectrometry utilized information-dependent acquisition (IDA) in high sensitivity mode, with a Declustering Potential (DP) of ±60 V (positive and negative modes) and Collision Energy of 35 ± 15 eV. IDA settings included excluding isotopes within 4 Da and monitoring 10 candidate ions per cycle. Data acquisition was segmented according to the mass range into four segments(50–300, 290–600, 590–900, and 890–1200,) thereby expanding the acquisition rate of secondary spectra. In each segment, four replicates were collected for each method. The acquired data were subjected to metabolite structure identification using the self-developed MetDDA and LipDDA methods.

### 2.7. Data Processing, Metabolite Identification, and Statistical Analysis

The statistical significance of differences in serum and intestinal mucosal samples between healthy and diarrheal piglets was evaluated using the Student’s unpaired *t*-test. The results are shown as means ± standard error of the mean (SEM). The data were analyzed using SPSS software (version 2.1.0; IBM, Armonk, NY, USA). A *p*-value < 0.05 was deemed to be statistically significant, whereas a 0.05 ≤ *p*-value < 0.10 was regarded as indicative of a tendency.

The data underwent conversion to the mzXML format using ProteoWizard (version 3.0; Palo Alto, CA, USA). Subsequently, peak alignment, retention time correction, and peak area extraction were performed using the XCMS (version 4.0.0; Bioconductor, Boston, MA, USA) tool. The identification of metabolite structures was performed by matching their precise masses (<25 ppm) and conducting second-level spectral matching using a laboratory-created database. Following Pareto scaling preprocessing, a series of multidimensional statistical studies were conducted, which included unsupervised principal component analysis (PCA) and orthogonal partial least squares discriminant analysis (OPLS-DA). Metabolites were identified by conducting a single-dimensional analysis using Student’s *t*-test and fold change analysis. The variable importance in projection (VIP) score for the first principal component in the OPLS-DA was also calculated. A 0.05 < *p*-value < 0.1, FC (Fold Change) > 1.5, and VIP > 1 were accepted as the proof of differences, and a *p*-value < 0.05 meant the differences were significant. The KEGG databases were consulted for the purpose of conducting pathway enrichment analysis.

## 3. Results

### 3.1. Serum Parameters of Weaning Stress and Proinflammatory Cytokines

Serum biochemical parameters of C-reactive protein and cortisol were determined (Figure 1a). Compared with those in the CT group, the concentrations of C-reactive protein and cortisol in the DIA group were significantly increased (*p* < 0.05). Then, serum proinflammatory cytokines were determined (Figure 1b). However, serum interleukin 18 (IL-18), IL-β, and tumor necrosis factor α (TNF-α) protein levels of the DIA group showed no significant changes (*p* > 0.05) compared with those in the CT group (Figure 1b).

### 3.2. The mRNA Expression of Intestinal Proinflammatory Cytokines

The mRNA expression of proinflammatory cytokines in the jejunum and colon were assessed (Figure 2). In jejunum, *IL-1β*, *IFN-γ*, and *TNF-α* mRNA level groups were significantly increased (*p* < 0.05) in the DIA group (Figure 2a) compared to those in the CT group. In the colon, the mRNA expression of *TNF-α* and *IFN-γ* were markedly increased (*p* < 0.05) in the DIA group compared to those in the CT group (Figure 2b).

### 3.3. The mRNA Expression of Tight Junction-Related Factors

In the jejunum (Figure 3a), the mRNA expression of zonula occludens-1 (*ZO-1*), *ZO-2*, occludin (*OCLN*), and claudin 1 (*CLDN1*) were markedly decreased (*p* < 0.05) in the DIA group compared to those in the CT group. In the colon (Figure 3b), the levels of *ZO-1*, *ZO-2*, and *CLDN1* mRNA were markedly decreased in the DIA group compared to those in the CT group.

### 3.4. LC-MS Analysis

Figure 4a,c depicts the PLS-DA model employed for dimensionality reduction analysis. A PLS-DA model was derived, where R^2^X = 0.31 corresponds to positive mode ionization and R^2^X = 0.34 represents negative mode ionization. To minimize the variations within each group and enhance the distinctions between the DIA and CT groups, we further implemented an OPLS-DA analysis. The model parameters for positive mode ionization were R^2^X = 0.44, while for negative mode ionization they were R^2^X = 0.43 (Figure 4b,d), and Q2 parameters in the positive mode (0.62) and negative mode (0.51) exceeded 0.5. The urine metabolites were plotted on a score plot, which clearly showed their separation inside the 95% Hotelling’s T2 ellipse. The results suggested that these models exhibit a minimal possibility of overfitting, along with commendable discrimination and prediction rates.

### 3.5. Metabolite Content Change and Metabolic Pathway Analysis

Initially, ninety-one metabolites (50 in the positive mode and 41 in the negative mode) were found (Appendix A). Then, using the *Kyoto Encyclopedia of Genes and Genome* (KEGG), pathway enrichment map analysis of the various metabolites in urine between the CT and DIA groups was performed. The differential metabolites between CT and DIA groups were shown in Figure 5. Purine metabolism and amino sugar and nucleotide sugar metabolism, which were among the top three significantly differential pathway, were specifically associated with urine. The differential metabolites (in positive and negative modes) involved in Purine metabolism and the amino sugar and nucleotide sugar metabolism pathway are shown in Figure 6. And, the locations of these metabolites within the pathway were depicted in Figure 7. The intensities of four metabolites (Deoxyinosine, Hypoxanthine, Inosine, and Guanosine) related to the hexosamine biosynthetic pathway and four metabolites (Glucosamine, N-Acetylmannosamine, Glucosamine-1-P, and N,N’-Diacetylchitobiose) related to the Purine metabolism were significantly increased (*p* < 0.05) in the DIA group compared to those in the CT group.

## 4. Discussion

In intensive pig farming, farmers usually adopt the strategy of early weaning, which involves weaning piglets at 21 to 28 days of age, to improve farming efficiency. Early weaning exacerbates weaning stress in piglets, resulting in diarrhea, increased mortality, and impaired growth performance. In the context of early weaning, this study found that piglets with post-weaning diarrhea experience higher stress levels than healthy piglets, shown by higher concentrations of C-reactive protein and cortisol. However, we did not observe any significant differences in TNF-α, IFN-γ, and IL-18 levels between the serum of diarrheal and healthy piglets. Normal proinflammatory cytokine levels might suggest that diarrheal piglets in our study were not experiencing severe systemic inflammation in a low-antigenic environment. Additionally, a previous study has reported that only 0.1% of blood cells are immune cells, constituting merely 1% of the total immune cells [22], even though blood comprises 90% of the cells of the body [23]. Therefore, we believed serum proinflammatory cytokines were not the most prominent and intuitive differences between diarrheal and healthy weaned piglets. In contrast, inflammatory and histological changes in early-weaned piglets primarily occur in the jejunum [24], with the colon playing an essential role in immune functions within the intestine. Our study showed that the mRNA expression of *TNF-α*, *IFN-γ*, and *IL-1β* was significantly overexpressed, and *ZO-1*, *ZO-2*, *OCLN*, and *CLND1* mRNA expression significantly decreased in the jejunum of diarrheal weaned piglets compared to healthy weaned piglets. Similarly, the colon of diarrheal weaned piglets exhibited dysregulation in the expression of immune-related and tight-junction-related factors. Intestinal tight junctions regulate the integrity of the intestinal physical barrier, which is essential for limiting the infiltration of pathogens into the lamina propria and aiding in the transportation of nutrients [25,26]. Our study noticed a disruption in the intestinal tight junctions in diarrheal weaned piglets. This disruption was probably caused by the considerable stress experienced throughout the weaning process. Prior studies have shown that weaning stress can disturb the integrity of intestinal tight junctions via the MAPK and TGF-β signaling pathways [5,27,28,29], resulting in an elevation in intestinal permeability. The high intestinal barrier permeability leads to the bypass of numerous antigen molecules through the paracellular route, contacting the basement membrane and ultimately triggering enteric infections [5,30]. In our experiment, piglets with suppressed intestinal tight junction expression exhibited proinflammatory cytokine overexpression, indicating intestinal inflammation. The increased secretion of inflammatory factors can further compromise the intestinal barrier [4,31], creating a vicious cycle that escalates the risk of gastrointestinal diseases in piglets. Ultimately, this leads to elevated diarrhea rates and reduced growth performance. In order to delve deeper into the metabolic factors contributing to inflammatory diseases in diarrheal weaned piglets, a metabolomic analysis was conducted to compare the distinct metabolites present in the urine of diarrheal and healthy weaned piglets.

This study found that the intensity of hexosamine biosynthetic pathway (HBP)-related metabolites was elevated in the urine of diarrheal weaned piglets compared to healthy weaned piglets. HBP represents a distinct nutritional sensing metabolic pathway, aside from glycolysis, involving glucose metabolism [32,33]. The ultimate product of HBP is UDP-N-acetylglucosamine (UDP-N-GlcNAc), which can subsequently be converted to O-GlcNAc by O-GlcNAc transferase, used for post-translational modification of various proteins [32,34,35]. Recent research has shown increased O-GlcNAcylation levels under stressful conditions [36,37]. Our study found a notable increase in metabolites associated with the HBP in piglets exposed to high weaning stress, which aligns with previous research results. Numerous studies have revealed that the HBP is a nutritional sensor in the inflammatory process [38,39,40,41,42]. According to a published study, the effect of O-GlcNAcylation on inflammation is contingent upon the glucose concentration [43]. GlcN promotes LPS-induced macrophage inflammation under normal glucose conditions in a dose-dependent manner. Conversely, GlcN suppresses macrophage inflammation at elevated glucose levels [43]. Puchalska et al. [44] further elucidated the inflammatory role of HBP under different energy metabolism conditions through isotope tracing analysis. The study indicated that HBP is involved in the polarization of both M1 and M2 macrophages, but the integration of intact mitochondrial and glucose metabolism with cytoplasmic HBP specifically participates in IL-4-induced M2 macrophage polarization [44]. This suggests that the connection between the HBP and inflammation depends on the proper functioning of mitochondrial energy metabolism. In this study, diarrheal weaned piglets exhibited an elevated expression of intestinal proinflammatory cytokines, concomitant with a significant increase in HBP-related metabolites. Considering that disrupted energy metabolism may synergistically drive inflammation through HBP and proinflammatory cytokines, the results of this study further speculate that diarrheal weaned piglets might experience disruptions in energy metabolism.

Given that the liver plays a crucial role in nutrition metabolism, disruptions in energy metabolism in diarrheal weaned piglets could be associated with hepatic dysfunction. A previous study reported that post-weaning piglets are at risk of developing hepatic inflammation [45]. Published studies have shown that individuals with hepatic inflammation have lower appetites [46] and higher energy expenditures [47,48]. A previous study has demonstrated that approximately 8% of the liver is occupied by immune cells [22]. The liver serves as the essential immune barrier, primarily countering continuous exposure to foreign antigens from the intestine [49]. This study revealed disrupted intestinal tight junctions in diarrheal weaned piglets. As mentioned earlier, the disruption of tight junctions would lead to the permeation of antigen molecules into other tissues [5,6]. Consequently, the results indicated that diarrheal weaned piglets were at a higher risk of developing hepatic inflammation. The liver is the primary site for purine metabolism. This study observed a notable rise in metabolites associated with the purine metabolic pathway in diarrheal weaned piglets, suggesting hepatic dysfunction. The hepatic inflammation undoubtedly exacerbates the nutritional burdens, especially the energy burden, on weaned piglets. These nutritional burdens are the reasons for the decreased growth performance. Furthermore, aberrations in purine metabolism can also be evidence of disrupted energy metabolism in diarrheal weaned piglets. Purine nucleotides also play a critical role in energy maintenance during the repairment of intestinal epithelial mucosa [50]. According to a study [51], at low purine levels, the intermediates of glycolysis are more utilized for serine synthesis. The diversion of glycolysis undoubtedly reduced the provision of energy. Likewise, prior studies have shown that purine metabolism disorders occur in the context of electron transport chain (ECT) dysfunction [52,53]. Therefore, the results of this study revealed abnormalities in purine metabolism in diarrheal weaned piglets, reiterating the disturbances in energy metabolism in diarrheal piglets.

Published research has reported that urate is the final product of purine metabolism, predominantly synthesized by tissues such as the liver via purine oxidase activity [54]. Approximately two-thirds of urate is excreted from the kidneys [55]. In this study, the metabolomic analysis showed a heightened intensity of urate in the urine of diarrheal piglets compared to controls, identifying it as a differential metabolite between diarrheal and healthy weaned piglets. A study has revealed that urea is a biomarker for oxidative stress in the body [56]. In the inflammatory process, both urate monosodium and soluble urate can induce the production of IL-1β by activating NLRP3 [57]. Thus, the elevated urate indicated inflammatory risks in the kidneys of diarrheal weaned piglets. Furthermore, abnormalities in uric acid metabolism led to urate buildup in the joints, ultimately causing lameness in pigs. According to statistics, lameness affects 0.6~2.2% of growing and fattening pigs [58,59,60]. While elevated urate levels causing lameness does not directly result in mortality, it is essential to note that pigs experiencing lameness suffer a significant decline in competitiveness over their prolonged growth phase. The decline ultimately causes pigs to be culled or euthanized and eliminated [60]. Our study revealed an increased urate level in the urine of piglets with post-weaning diarrhea. This could significantly contribute to diminished growth performance and increased mortality in diarrheal weaned piglets.

## 5. Conclusions

In conclusion, this experiment observed intestinal inflammation in weaned piglets despite the absence of significant changes in serum inflammatory factors. The results demonstrated significant alterations in purine metabolism and HBP between diarrheal and healthy weaned piglets within this situation. Furthermore, purine metabolism and HBP abnormalities indicated energy metabolism-related diseases in diarrheal weaned piglets. The results of this study suggest that further studies could focus on dietary strategies, including raising feed energy levels and anti-inflammatory nutrients to reduce mitochondrial energy metabolism disorders and ultimately alleviate diarrhea in weaning piglets.

## Figures and Tables

**Figure 1 animals-14-00522-f001:**
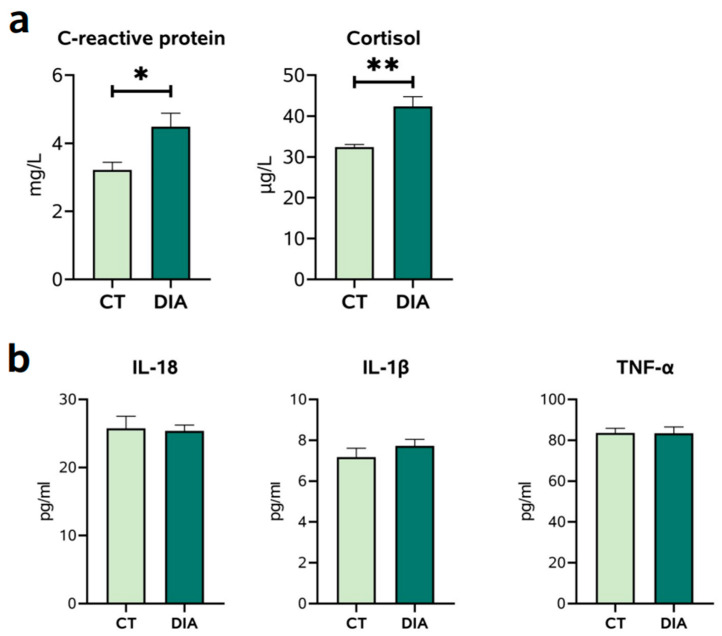
(**a**) Serum C-reactive protein and cortisol levels. (**b**) Serum interleukin 18 (IL-18), 1β (IL-1β), and tumor necrosis factor α (TNF-α) protein levels. Results are expressed as means ± SEM (*n* = 9), and significant differences between control (CT) and diarrhea (DIA) groups are indicated by * 0.01 < *p* < 0.05; ** 0.001< *p* < 0.01.

**Figure 2 animals-14-00522-f002:**
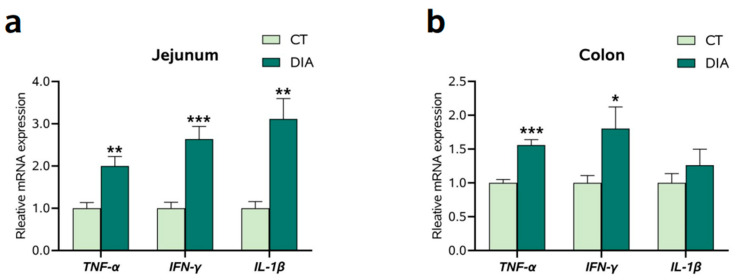
The relative expression of interleukin 1β (*IL-1β*), interferon γ (*IFN-γ*), and tumor necrosis factor α (*TNF-α*): (**a**) Jejunum; (**b**) Colon. Results are expressed as means ± SEM (*n* = 9), and significant differences between the control (CT) and diarrhea (DIA) groups are indicated by * 0.01 < *p* < 0.05; ** 0.001 < *p* < 0.01; and *** *p* < 0.001.

**Figure 3 animals-14-00522-f003:**
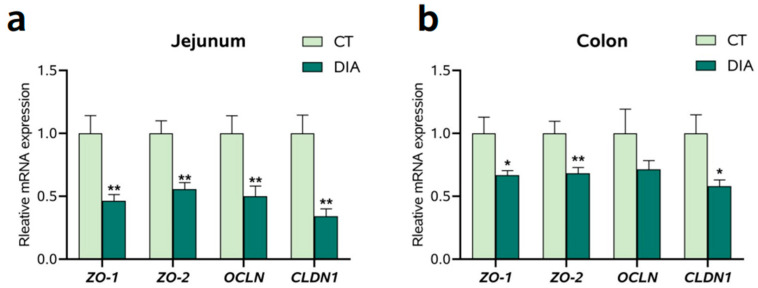
The relative expression of zonula occludens-1 (*ZO-1*), *ZO-2*, occludin (*OCLN*), and claudin 1 (*CLDN1*): (**a**) Jejunum; (**b**) Colon. Results are expressed as means ± SEM (*n* = 9), and significant differences between the control (CT) and the diarrhea (DIA) groups are indicated by * 0.01 < *p* < 0.05; ** 0.001 < *p* < 0.01.

**Figure 4 animals-14-00522-f004:**
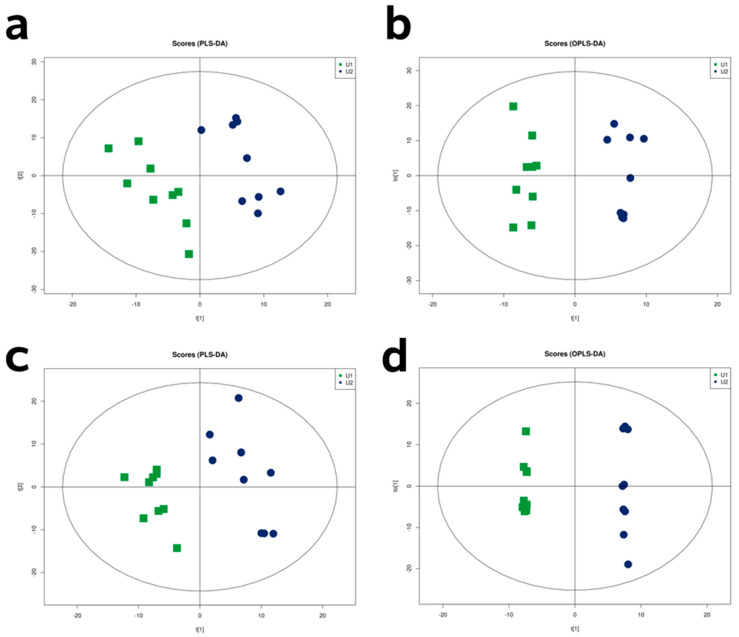
Partial least squares discriminant analysis (PLS-DA) and orthogonal partial least squares discriminant analysis (OPLS-DA) two-dimensional score plots of urine metabolites in comparisons to the control (CT) and diarrhea (DIA) groups following positive (**a**,**b**) and negative (**c**,**d**) mode ionization. In the plots, U1 represents the CT group and U2 represents the DIA group.

**Figure 5 animals-14-00522-f005:**
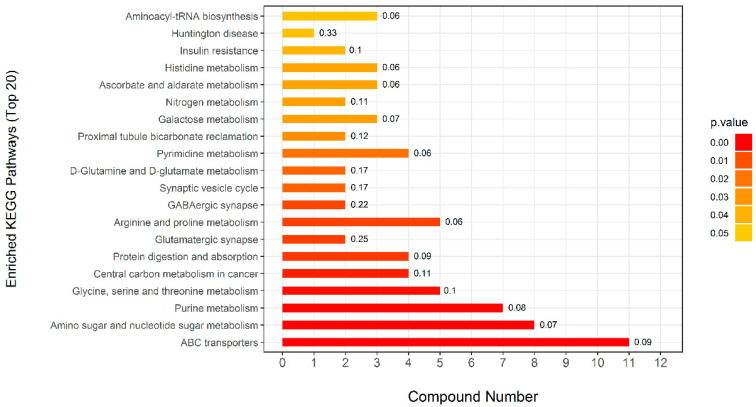
Pathway enrichment map analysis of differential metabolites in urine between the control (CT) and diarrhea (DIA) groups using the *Kyoto Encyclopedia of Genes and Genomes* (KEGG).

**Figure 6 animals-14-00522-f006:**
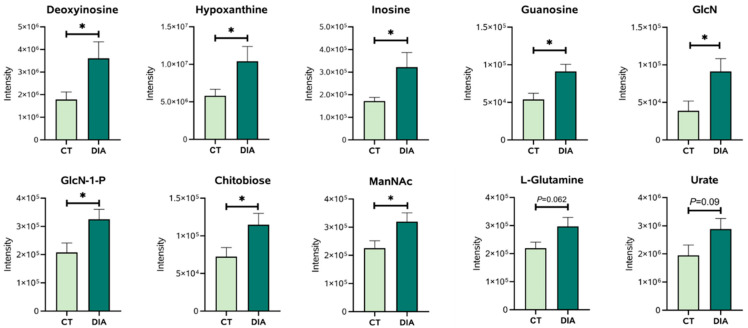
Intensities of metabolites identified in the positive and negative modes. CT, control group; DIA, diarrhea group; GlcN, Glucosamine; ManNAc, N-Acetylmannosamine. Results are expressed as means ± SEM (*n* = 9), and significant differences between the CT and DIA groups are indicated by * 0.01 < *p* < 0.05.

**Figure 7 animals-14-00522-f007:**
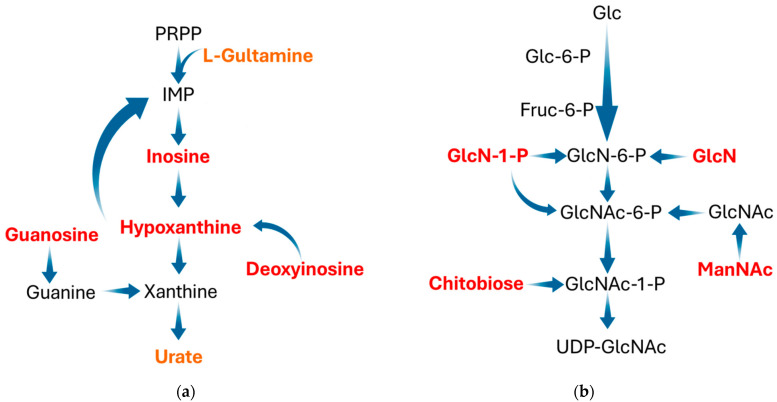
Labeling map depicting integration of purine metabolism (**a**) and hexosamine biosynthetic pathway (**b**). Significantly increased metabolites in urine of diarrheal weaned piglets are indicated by red text, and marginally increased metabolites are indicated in orange. PRPP, Phosphoribosyl pyrophosphate; IMP, Inosine Monophosphate; Glc, Glucose; GlcN, Glucosamine; GlcNAc, N-Acetylglucosamine; ManNAc, N-Acetylmannosamine; Fruc, Fructose.

**Table 1 animals-14-00522-t001:** Sequence of primers used for the real-time quantitative PCR analysis.

Genes ^1^	Primer and Probe Sequences (5′-3′) ^2^	Product Length/bp	Annealing Temperature (°C)
*TNF-α*	F: CTACTCGTCCAACGGGAAAG	121	59.7
	R: ACGCCTCCAAGTTACCACTG		
*IL-1β*	F: CAGCCCCCGTACATGGAGA	112	59.7
	R: GCGCAGACGGTGTTCATAGTT		
*IFN-γ*	F: GCATCATTTCCTCCCTGTT	155	59.7
	R: TCTTGGCTTTGGGTGGTT		
*ZO-1*	F: CTGAGGGAATTGGGCAGGAA	147	59.7
	R: TCACCAAAGGACTCAGCAGG		
*ZO-2*	F: ATTCGGACCCATAGCAGACATAG	126	59.7
	R: GCGTCTCTTGGTTCTGTTTTAGC		
*OCLN*	F: CAGGTGCACCCTCCAGATTG	149	59.7
	R: GGACTTTCAAGAGGCCTGGAT		
*CLDN-1*	F: AGTAGGGCACCTCCCAGAAG	137	59.7
	R: CCTGAACTCCCTCTACTTGTGTTC		
*β-actin*	F: TCTGGCACCACACCTTCT	114	59.7
	R: TGATCTGGGTCATCTTCTCAC		

^1^ *TNF-α =* tumor necrosis factor α, *IL-1β =* interleukin 1β, *IFN-γ =* interferon γ, *ZO-1 =* zonula occludens-1, *ZO-2 =* zonula occludens-2, *OCLN =* occluding, *CLDN-1 =* claudin 1. ^2^ F = forward primer; R = reverse primer.

## Data Availability

The data presented in this study are available on request from the corresponding author. The data are not publicly available due to the protection of data privacy and intellectual property rights.

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
