# Peer review of "Purine Metabolism and Hexosamine Biosynthetic Pathway Abnormalities in Diarrheal Weaned Piglets Identified Using Metabolomics"

_animals, 2024, doi:10.3390/ani14030522_

Round 1

Reviewer 1 Report

Comments and Suggestions for Authors

The issue concerning post-weaning diarrhea seriously influences the pig production. The authors conducted this study to investigate the intestinal injury of diarrheal weaned piglets, and identify the differential metabolites in the urine of diarrheal weaned piglets and those with healthy weaned piglets. This study observed that the elevated weaning stress and inflammatory disease were associated with the abnormalities of purine metabolism and hexosamine biosynthetic pathway of weaning piglets, which may provide important information to the pig production. The experimental design and whole presentation were fine. However, there are some issues need to be solved by the authors.

1. The authors should provide the information of weaned piglets in the abstract section, such as age and weight.

2. The standard for diarrhea scoring should be presented in the M&M section as a sentence instead of in a Table. Additional, please provide the reason how to judge the piglets with diarrhea scores above 16 as diarrhea. I got confused, considering that the maximum score is 3, how to get 16 within 4 days?

3. The information of body weight should be presented in the abstract and text.

4. It would be better to move Tables S3 and S4 in the main text.

5. The authors should provide the method how to collect the urine sample.

6. There are many formative and grammatical issues in the text that should be carefully modified by the authors. In addition, there are some typo, such as Line 78, “simple” should be “sample”; Line 246, “ou” should “our”; Line 248, “are” should be “were”. “weaning piglets” should be “weaned piglets”

7. The description for significance is wrong, should be * 0.01< P < 0.05; **0.001< P < 0.01; *** P < 0.001.

Comments on the Quality of English Language

There are many formative and grammatical issues in the text that should be carefully modified by the authors. In addition, there are some typo, such as Line 78, “simple” should be “sample”; Line 246, “ou” should “our”; Line 248, “are” should be “were”. “weaning piglets” should be “weaned piglets”.

Thus, a language editing is required. 

Reviewer 2 Report

Comments and Suggestions for Authors

While the study addresses the significant issue of post-weaning diarrhea in pig production and provides valuable insights into metabolic changes, there are areas for improvement. Firstly, the manuscript lacks clarity in presenting the research design and methods. The introduction provides a concise overview of the issue, but expanding it would offer a more comprehensive background for readers. In statistical analysis student t-test has been used, but not on all studied parameters, this test can be applied, therefore, it is better to clarify it. Detailed information on the selection criteria for piglets, experimental procedures, and statistical analyses would enhance the study's reproducibility and rigor. Additionally, the interpretation of results could be more comprehensive. Further discussion on the potential implications of elevated C-reactive protein and cortisol levels, as well as the significance of altered mRNA expression in inflammatory markers and tight junction proteins, would provide a deeper understanding of the observed metabolic changes. Furthermore, the metabolomic analysis findings need more contextualization to link the identified metabolites to specific pathways and their implications for the piglets' health. Addressing these aspects would strengthen the paper and contribute to its scientific impact.

Reviewer 3 Report

Comments and Suggestions for Authors

Purine metabolism and hexosamine biosynthetic pathway abnormalities in 2 weaned diarrhea piglets identified by metabolomics 1

This paper evaluates the metabolomics of urine samples from weaned piglets with diarrea vs health individuals. Authors also measured the oxidative stress of both groups of piglets and the expression of some indicators of gut health. The paper has some spelling mistakes and writting needs to be improved, mainly in the discussion section. Some details in the material and methods are required. Results are clearly presented and looks like attrative for readers.

Specific comments

Line 42-44 “The antibiotic residues and the antibiotic resistance pose significant threat to both human and animal  wellness [11, 12]. Currently, there exists a strong worldwide demand among consumers to minimize the utilization of antibiotics [13, 14].”.

This sentences seem to be of similar meaning and repetitive. Please improve writing and sum up the main idea concerning to antibiotic use. You can sum up these two sentences in one

Lines 55-57. “This study interprets the high stress and inflammatory disease in piglets with post-weaning diarrhea, employing a metabolic  framework. The findings of this study contribute significant insights for nutritional strat- egies aimed at alleviating post-weaning diarrea”

This seems to be more a conclusion than an objective. Please identify clearly the objetives of the study

Line 78 Simple or simple??

Line 95. Please indicate more details (model, provider and address) for the biochemical analyzer

Line 119. Liquid chromatography?. Which analysis did you performed?

Line 120 samples?, which samples? Urine?. How was the preparation of the samples prior to injection??. There is only information of the cromatografic conditions.

Lines 137-142 “Following Pareto scaling preprocessing, a series of multidimensional statistical studies were conducted, which included unsupervised principal component analysis (PCA) and orthogonal partial least squares discriminant analysis (OPLS-DA). The statistical studies performed were Student's t-test and fold change analysis, both of which were single-dimensional. The variable importance in pro-jection (VIP) score for the first principal component in the OPLS-DA was also calculated”

All this information could be moved to statistical analysis. Some information is repeted in the statistical analysis section. Please put together and explain it more clearly.

Discussion. Needs of writting improvement. There are repetitions, spelling mistakes and it is desorganized.  

Line 245. “serum of diarrheal piglets and healthy piglets.”. repeated “piglets”

Line 246. “ou” or our?

Line 272 pervious or previous??

Lines 254-260. “Our study showed that the mRNA expression of TNF-α, IFN-γ, and IL-1β 254 significantly overexpressed, and ZO-1, ZO-2, OCLN, and CLND1 mRNA expression were significantly suppressed i……”

Please discuss this data and look for other studies concerning the expression of these values. For example the study of “Laviano et al. (2023), also found lower OCLN expression in those groups with higher oxidative status. Laviano et al. Maternal Supplementation of Vitamin E or Its Combination with Hydroxytyrosol Increases the Gut Health and Short Chain Fatty Acids of Piglets at Weaning. Antioxidants 2023, 12, 1761. https://doi.org/ 10.3390/antiox12091761

Lines 264-the end. Authors need to organize better the discussion and avoid repetition of the same ideas. Discussion ideas should be linked. It is not a compilation of results that other investigators found. Please improve writing.

Lines 337-338. “Given the importance of urine as a vital excretory route for purine metabolites, our study focused on the examination of metabolites of urine. “

This idea should be at the beggining of the paragraph and not at the end.

Lines 328-336- @The present study revealed evidence of inflammation and abnormalities in purine- related metabolites in diarrheal piglets, although we not directly measured the levels of inflammatory cytokines of liver…..”

There is a wide explanation about inflamation in liver, but authors did not evaluate any parameter in liver. Please try to sum up what the authors want to say.

Authors must improve writting!

Comments on the Quality of English Language

Moderate editing of English required

Round 2

Reviewer 3 Report

Comments and Suggestions for Authors

Please check the following sentence:

"After slaughter, the urinary bladder of piglets is excised, and urine is  extracted from the urinary bladder using a syringe for metabolomic analysis"

You are writing in past tense so it should be better in past (was excised and is extracted instead is)

Comments on the Quality of English Language

Minor changes